# Insights from Real-World Practice: The Dynamics of SARS-CoV-2 Infections and Vaccinations in a Large German Multiple Sclerosis Cohort

**DOI:** 10.3390/vaccines12030265

**Published:** 2024-03-03

**Authors:** Hernan Inojosa, Dirk Schriefer, Yassin Atta, Anja Dillenseger, Undine Proschmann, Katharina Schleußner, Christina Woopen, Tjalf Ziemssen, Katja Akgün

**Affiliations:** Center of Clinical Neurosciences, Department of Neurology, University Hospital Carl Gustav Carus, Technical University of Dresden, 01307 Dresden, Germany; hernan.inojosa@ukdd.de (H.I.); dirk.schriefer@ukdd.de (D.S.); yassin.atta@ukdd.de (Y.A.); anja.dillenseger@ukdd.de (A.D.); undine.proschmann@ukdd.de (U.P.); katharina.schleussner@ukdd.de (K.S.); christina.woopen@ukdd.de (C.W.); tjalf.ziemssen@ukdd.de (T.Z.)

**Keywords:** multiple sclerosis, SARS-CoV-2, COVID-19, vaccinations, infection patterns, disease-modifying therapies

## Abstract

The SARS-CoV-2 pandemic profoundly impacted people with multiple sclerosis (pwMS). Disease-related aspects and demographic factors may influence vaccination rates, infection susceptibility, and severity. Despite prior research, comprehensive real-world data obtained throughout the pandemic remain limited. We investigated SARS-CoV-2 vaccination and infection patterns in a large monocentric real-world cohort. We collected prospective data from medical visits at the MS Center Dresden, Germany, from the pandemic’s beginning until 31 May 2022. Logistic regression and rank correlation analyses were used to explore associations between SARS-CoV-2 outcomes and patient characteristics. Of 2115 pwMS assessed (mean age 46.5, SD ± 12.9; median expanded disability status scale 2.5), 77.9% were under disease-modifying treatment (DMT), primarily B-cell depletion (25.4%). A total of 35.5% reported SARS-CoV-2 infections, and 77.4% were fully vaccinated. PwMS with increased disability, older age, and comorbidities were associated with higher vaccination rates, possibly due to the awareness of these populations regarding complications of SARS-CoV-2 infections. Infections were more common in younger females, people with a lower degree of disability, those with relapsing MS, and those who were not vaccinated. PwMS on B-cell depletion reported more infections than untreated pwMS and those receiving other types of disease-modifying therapy, despite higher vaccination rates. Most infections were mild, with no severity differences according to demographic or disease-related factors, except for gender. Notably, all fatal cases occurred in unvaccinated pwMS. Our studies suggest that demographic and disease-related factors, especially age and the use of B-cell depletion, significantly influenced SARS-CoV-2 vaccination and infection rates in our cohort. These factors may be considered in future preventive campaigns in further pandemics.

## 1. Introduction

The SARS-CoV-2 coronavirus strain rapidly spread across the globe, resulting in critical consequences for healthcare in recent years. This virus is responsible for causing various respiratory diseases, ranging from a mild common cold or bronchiolitis to severe pneumonia, respiratory distress, and death [1]. This has posed a significant challenge in the management of people with multiple sclerosis (pwMS), which is one of the most prevalent chronic autoimmune diseases of the central nervous system (CNS), with approximately 250,000 patients in Germany alone [2].

Several disease-related factors can affect the risk and severity of infection and vaccination rates in pwMS [3]. While mixed evidence is available, certain factors, such as the level of disability, demographic factors, or the use of disease-modifying therapies (DMTs), can play a significant role in MS [4,5,6,7,8]. Younger female patients seem to be at a higher risk of infection, whereas cardiovascular comorbidities, progressive MS forms, male sex, and increasing age are associated with higher mortality rates [9,10,11]. The effect of DMTs on the risk of infections and severity is still uncertain.

Moreover, patterns of vaccination, a demonstrated effective measure against the pandemic, can vary among pwMS. The vaccination willingness in these patients has been reported to be moderately high, similar to the general population [12,13]. However, not only can disease-related aspects affect these rates, but the use of certain DMTs [e.g., B-cell depletion or sphingosine-1-phosphate receptor (S1PR) modulators] may also cause impaired humoral and/or cellular responses after vaccine administration and therefore affect their efficacy [14,15,16,17].

In Germany, the World Health Organization (WHO) reports over 37 million confirmed cases of SARS-CoV-2 and 160,000 deaths occurring in several phases and waves. Regarding vaccinations, more than 192 million doses have been administered [18]. Understanding the dynamics of SARS-CoV-2 infections and vaccinations during the pandemic is crucial, particularly for specific groups like MS populations. Although several studies have examined the outcomes of SARS-CoV-2 infections and vaccinations in pwMS, as far as we know, an analysis of real-world clinical practice in a large clinical cohort covering a period that includes the main course of the pandemic is still lacking [19,20,21].

In this study, we aimed to assess a large real-world cohort of MS patients during the pandemic and evaluate the occurrence of infections and vaccinations according to several disease-specific, demographic, and temporal factors. This could provide important additional information for the development of specific preventive strategies for pwMS in the future.

## 2. Materials and Methods

We conducted a prospective, monocentric, observational study at the Multiple Sclerosis Center at the University Hospital Carl Gustav Carus, Dresden, Germany. Data were acquired from the beginning of the pandemic until 31 May 2022. We included patients who had a confirmed diagnosis of MS or clinically isolated syndrome (CIS) according to the 2017 revisions of the McDonald criteria [22], were older than 18 years, and had the ability to provide written informed consent. PwMS were classified into groups with relapsing–remitting MS (RRMS), secondary progressive MS (SPMS), and primary progressive MS (PPMS).

### 2.1. Data Collection

Data were collected by neurologists and neurology residents during routine medical visits every three months over the observation period. SARS-CoV-2 infections and administered vaccinations were registered by patient anamnesis.

The date and number of doses received were documented for the vaccinations. The following vaccines were reported and considered for analysis: Comirnaty^®^ by Pfizer/BioNTech Manufacturing GmbH (Mainz, Germany), Vaxzevria^®^ by Oxford/AstraZeneca AB (Nijmegen, The Netherlands), Jcovden^®^ by Janssen (Johnson & Johnson) (Beerse, Belgium), Spikevax^®^ by Moderna (Cambridge, MA, USA), Nuvaxovid^®^ by Novavax (Gaithersburg, MD, USA), and CoronaVac^®^ by Sinovac (Beijing, China).

Laboratory-confirmed SARS-CoV-2 infections, either in the form of reverse transcription polymerase chain reaction (RT-PCR) or a rapid antigen test (RAT), were collected continuously during the study period. The severity of infections was established by the treating neurologists as follows: asymptomatic; mild (mild common cold symptoms and/or low-grade fever); moderate (hospitalization, intense symptoms without critical care management, high-grade fever, fatigue, pneumonia); severe (hospitalization with respiratory distress symptoms, critical care management); and death due to infection and complications. In the case of additional information reported by patients between on-site appointments (e.g., via email and telephone), this was confirmed and documented in the immediately following medical visit. Moreover, pwMS were asked to provide a current negative RT-PCR or RAT before they visited the center (performed within the last 72 or 24 h, respectively) as part of preventive measures sustained over the observation period.

Comorbidities were collected by a retrospective electronic medical chart review. These included the presence of cardiovascular diseases, obesity, diabetes, cancer, chronic kidney disease, and chronic obstructive pulmonary disease, among others.

Disability status was assessed regularly with the expanded disability status scale (EDSS) [23]. DMTs were analyzed according to the mechanism of action as platform injectables (interferon and glatiramer acetate), platform oral (dimethyl fumarate, diroximel fumarate, and teriflunomide), S1PR modulation (fingolimod, ozanimod, siponimod, and ponesimod), B-cell depletion (ocrelizumab, ofatumumab, and rituximab), induction therapies (alemtuzumab and cladribine), VCAM-1 blocker (natalizumab), and others (e.g., in the setting of a clinical trial, azathioprine).

For a stratified analysis of incident events (vaccinations and infections) over time, we followed the retrospective division of the pandemic into eight phases according to the wave classification of the Robert Koch Institute (RKI), Germany’s national public health agency: sporadic infections before wave 1, waves 1 to 5, and summer plateaus 2020 and 2021 (Table 1) [24]. This classification is based on various epidemiological and healthcare- and policy-related metrics [25].

### 2.2. Statistical Analysis

Quantitative measures are summarized as means and standard deviations (SDs) or medians and interquartile ranges (IQRs). Categorical measures and patient subgroups are summarized as absolute (counts) and relative frequencies (percentages). For the SARS-CoV-2-related outcomes of primary interest, 100% stacked bar charts are used to fully visualize the distribution of relative frequencies across each outcome category: (i) number of vaccinations against SARS-CoV-2 (0, 1, 2, 3, 4, 5+), (ii) number of SARS-CoV-2 infections (0, 1, 2), (iii) severity of infections (asymptomatic, mild, moderate, severe, death). The graphical bar chart representation was implemented for the overall cohort as well as for specific subgroups, as defined by sociodemographic (e.g., age, sex) and disease-related characteristics (e.g., DMT, EDSS, presence of comorbidities) and the eight temporal phases of the pandemic (Table 2). For inferential analysis, a dichotomous quantification was performed using the following cut-offs, which have been considered particularly meaningful [26]: (i) vaccination status: at least two (≥2) versus fewer doses (0–1); (ii) presence of a SARS-CoV-2 infection: never (0) versus ever (1+); (iii) severity of infection: non-mild (moderate, severe, death) versus mild (asymptomatic, mild). To investigate cross-sectional associations between these binary dependent variables and sociodemographic as well as disease-related independent variables, univariable and multivariable logistic regression models were generated. Given the role of age and sex as potential confounders [2,3], multivariate analyses were adjusted for both age and sex. Exponentiated slope coefficients (odds ratios) and associated 95% confidence intervals are reported. When dealing with categorical predictors (independent variables), odds ratios are presented in relation to a specified reference category. For pairwise comparisons between the eight DMT groups, the Bonferroni adjustment was used to control for the type I error rate. To account for the global-scale level of the outcomes (i.e., without the use of categorizations for ordinal-scaled SARS-CoV-2 outcomes), Spearman rank correlations were calculated to represent associated (bivariate) interrelationships. As a conservative benchmark, correlations of 0.10, 0.30, and 0.50 were considered relatively small, medium, and large effect sizes, respectively. The longitudinal analysis involved a descriptive representation of the primary outcomes across the temporal phases of the pandemic. For all inferential analyses, a *p*-value < 0.05 was considered a criterion of statistical significance. Calculations were performed using IBM SPSS version 28 (IBM Corporation, Armonk, NY, USA), and figures were created with GraphPad Prism version 5 (GraphPad Software Inc., La Jolla, CA, USA).

The study complies with the Declaration of Helsinki and was approved by the local ethics committee of the University Hospital Dresden (BO-EK-329072022). Patients provided their written informed consent.

## 3. Results

### 3.1. Patient Characteristics

A total of 2115 pwMS were eligible for analysis. The mean age was 46.5 ± 12.9 years, and three out of four study participants were female (72.5%) (Table 2). Most pwMS were diagnosed with RRMS (80%), and more than half presented with at least one comorbidity (62.4%). The median EDSS was 2.5 points (IQR 1.5–4.0), and the median disease duration was 10 years (IQR 5–16). The most frequently administered DMT was B-cell depletion (25.4%), followed by S1PR modulation (15.9%) and platform DMT in either oral (12.6%) or injectable form (10.9%). In contrast, approximately one-fifth (21.6%) received no DMT (Table 2, Appendix A).

### 3.2. Vaccine- and Infection-Related Characteristics

In nearly two-thirds of the study population, no confirmed SARS-CoV-2 infections occurred (64.5%). The mean number of infections was 0.39 ± 0.55, with 32.3% of pwMS experiencing one infection and 3.1% experiencing two infections during the observation period. In cases with the documented severity of infection, these were most frequently mild (Table 3). Two or more vaccinations were administered in 77.4%, and three or more vaccinations were documented in 62% of pwMS. A status of either two vaccinations or one infection (recovery) plus one vaccination was achieved in 80.6% of pwMS. Three or more incidents of vaccination or infection events were identified in 71.6% of the group (Table 3). Conversely, in one in ten pwMS, neither a SARS-CoV-2 vaccination nor infection was reported (9.8%).

Overall, pwMS were most frequently vaccinated with Comirnaty^®^ (81.5% of a total of 4778 documented doses), followed by Spikevax^®^ (11.3%), Vaxzevria^®^ (3.3%), and Nuvaxovid^®^ (2.4%). The number of vaccinations per patient observed in different patient subgroups is given in Figure 1a, while the number of infections in these subgroups is shown in Figure 1b.

### 3.3. Vaccination Patterns According to Demographic and Clinical Characteristics

The proportion of pwMS having received at least two doses of SARS-CoV-2 vaccines was significantly higher in older pwMS, pwMS presenting with comorbidities, and those with more severe disability in comparison with younger, disability-free, and low-disability subgroups (all *p* < 0.05; Figure 1a, Table 4 and Table 5). In contrast, the proportion of pwMS with two or more vaccinations did not differ significantly by sex or the clinical course. In this comparison, subjects treated with B-cell depletion therapies were in the subgroup of patients with the highest rate of two or more vaccinations, especially when compared to patients without therapy or with platform injectable DMTs (both *p* < 0.05; Figure 2, Appendix A). No significant differences in the frequency of vaccination were observed regarding clinical courses. The results of logistic regression models are shown in detail in Table 4. After multivariable adjustment, age (years) and the use of B-cell depletion therapies remained consistently significantly associated with increased odds of having received at least two vaccinations [OR 1.016, 95% CI (1.008–1.024) and 1.938, 95% CI (1.429–2.629), respectively].

### 3.4. SARS-CoV-2 Infections According to Demographic and Clinical Characteristics

A higher proportion of SARS-CoV-2 infections was observed for female, younger pwMS and for pwMS without comorbidities compared to their male and older counterparts, as well as pwMS with comorbidities (all *p* < 0.05, Figure 1b, Table 4 and Table 5). Similarly, a significantly higher proportion of infections was seen in pwMS presenting with a relapsing MS course and a low level of disability. Further, infections were significantly more frequent in pwMS with B-cell depletion therapies compared to those without DMT. After multivariable adjustment, older age, male sex, and higher EDSS remained associated with lower odds of having experienced SARS-CoV-2 infections, and the use of B-cell depletion therapies with higher odds of having experienced SARS-CoV-2 infections (Table 4).

Correlation analysis revealed consistent negative correlation patterns for patient characteristics with the number of infections (r < 0) and consistent positive correlation patterns with the number of vaccinations (r > 0) (Table 5). The associated effect sizes were relatively low (range r = 0.057 to r = 0.150), with age being the most striking, followed by EDSS. The strongest cross-sectional correlation was found for the number of infections and number of vaccinations (moderate effect size of r = −0.314). This indicates an inverse relationship in which a higher number of infections is (monotonically) associated with a lower number of vaccinations and vice versa (Table 5).

Likewise, a significantly higher proportion of non-vaccinated pwMS experienced an infection compared to those pwMS having received two, three, or more doses of the vaccine [30% vs. 54.3%; OR 2.78 (95% CI 2.25–3.42)].

### 3.5. Severity of SARS-CoV-2 Infections

In our study, we found that the majority of documented SARS-CoV-2 cases had a mild disease course (61.8%), while only a few cases were asymptomatic (6.4%). Fatal and severe cases were rare in our cohort (Table 3, Figure 2). Male patients had a higher proportion of moderate/severe cases (*p* = 0.029). We did not find any significant associations between disease severity and patient subgroups defined by age, comorbidities, disability, MS course, use of DMT, or vaccination status (Figure 2, Appendix A). However, fatal cases due to SARS-CoV-2 were only observed in unvaccinated pwMS (n = 4).

### 3.6. Temporal Distribution of Infections and Vaccinations across the Different Phases of the Pandemic

Considering the different phases of the pandemic (Table 1), most of the SARS-CoV-2 cases were reported during the fifth wave (18.7%), which was dominated by the Omicron variant of concern (as depicted in Figure 3a). Only a few cases were reported during the early stages of the pandemic, with just one case during the sporadic infection phase and three cases in the first wave of SARS-CoV-2.

Out of all pwMS, 2.6% reported receiving two or more vaccinations during the second wave of the pandemic. This number rapidly increased during the third wave as vaccinations became widely available. By the end of the fifth wave, 77% of pwMS had received at least two doses of the vaccine. During the second wave, a slightly higher proportion of unvaccinated pwMS reported infections. This difference became more pronounced as the pandemic progressed, with the fifth wave showing the highest relative difference (Figure 3a).

The dynamics of both vaccinations and recoveries from infections are shown in Figure 3b. With the second wave, there is a noticeable increase in both cumulative recoveries and vaccinations. In the third wave, more than 60% of pwMS had been vaccinated at least once or had had a SARS-CoV-2 infection. At the end of the observation period, approximately 80% of the group had experienced at least two of these immunological events. As of the end of 2021, German authorities recommended a booster vaccination, at least three months after the second vaccination. This was performed by 39.9% and 56.9% of pwMS during the fourth (Delta) and fifth (Omicron) waves, respectively.

## 4. Discussion

We conducted a comprehensive analysis of vaccination and infection patterns during the SARS-CoV-2 pandemic in a large German cohort of pwMS. This study provides valuable insights into demographic and disease-specific factors that impacted the pandemic’s course in this population. Our study cohort consisted mostly of young female patients diagnosed with relapsing–remitting MS, with a median EDSS score of 2.5. This demographic and clinical profile is commonly observed in MS populations [2].

As a regional reference center, most patients received B-cell depletion as a form of DMT, which may differ from other clinical settings. Regarding SARS-CoV-2 vaccinations, 77.4% of people with multiple sclerosis (pwMS) had received two or more doses, and more than 80% of the group had experienced at least two immunological events (i.e., recoveries or vaccinations) by the end of the observation period. A group of pwMS were vaccinated in the second wave, shortly after the availability of SARS-CoV-2 vaccinations in Germany. The proportions of vaccinated pwMS were similar to the general German population at this time and by the end of the observation period in the fifth wave (2.55% vs. 2.5% and 77% vs. 75.9%, respectively) [26]. Similar findings were observed in an Australian cohort [17].

It is interesting to note that there were significant differences in vaccination rates among pwMS based on their age and disability level. Older age, the presence of comorbidities, and more severe disability, as assessed with the EDSS, were associated with higher immunization rates. This could be because populations with these characteristics may be more aware of the possible risks and severe course of SARS-CoV-2 infections, and vaccination campaigns were more aggressively targeted toward people with risk factors for a severe disease course, regardless of their MS diagnosis [17,27,28]. Furthermore, pwMS with B-cell depletion had a higher proportion of vaccinations compared to other DMT groups. Vaccinations under this DMT are usually discussed in detail during DMT selection due to the mechanism of action of these drugs. As a lower humoral immune response is described for B-cell depletion therapies, efforts for immunization are directed more intensely toward these pwMS [29]. In our cohort, we could not confirm previously reported higher hesitancy among females, although regional differences should be accounted for [17].

About one-third (35.5%) of the group reported having been infected with SARS-CoV-2, while a considerable proportion, almost two-thirds, of pwMS remained uninfected. This compares to a total of 31,701 recoveries per 100,000 persons, as estimated by the Robert Koch Institute for Germany, by the end of the observation period [26]. Therefore, the incidence in our cohort can be considered similar to that in the general population. We also noted that younger female subjects without comorbidities and with relapsing MS with lower disability scores had reported more infections than those with different characteristics. As populations with these characteristics were demonstrated to have a lower risk of a severe disease course, it is possible that preventive measures were taken with a lower intensity in this subgroup of pwMS. It is likewise possible that this population may have had a more active participation in social and professional life during the pandemic, disregarding quarantine recommendations or preventive measures and therefore presenting a higher risk of infection [30,31,32].

The relationship between SARS-CoV-2 and DMTs has been widely discussed since the beginning of the pandemic. At our center, a higher proportion of pwMS treated with B-cell depletion reported infections compared to patients in other categories. However, this population was usually vaccinated more often than their counterparts. B-cell depletion therapies may impair the humoral response and the production of new antibodies, leading to an impaired immune response to infections and immunizations [33]. Nevertheless, the cellular immune response in this group is present and, according to several reports, even superior to that in pwMS receiving no DMT or S1PR modulation [16,32,34]. However, the higher proportion of patients with B-cell depletion in our cohort may have contributed to the significance of these results. Due to the observational real-world characteristics of our study, these results should be interpreted carefully. For instance, a large case–control study showed a higher risk of SARS-CoV-2 in patients treated with natalizumab, suggesting a possible role of frequent hospital visits in pwMS with this DMT [9].

Moreover, pwMS without DMT had relatively few infections, with a similar tendency observed for patients on platform injectable or induction DMTs, even though no significance was observed in the latest, possibly due to the proportional size of both groups. The effect of DMTs on the acquisition of SARS-CoV-2 infections is still unclear [35,36,37]. In certain contexts, MS therapies were even postulated as possible tools to alter the disease course and reduce disease severity [38,39].

As shown by the correlation and regression analyses, age appears to be the most important factor with the strongest associations with the number of vaccinations and infections. These findings further suggest that age may act as a confounder in the relationship between independent variables (such as number of comorbidities, type of MS, and disease duration) and these pandemic-related dependent variables, thus influencing the associations observed in the univariate analysis. In general, vaccinated pwMS reported less frequent SARS-CoV-2 infections than the unvaccinated. Conversely, no association was observed between the use of vaccinations and the disease course. Only sex seemed to play a significant role in our cohort with regard to the severity of infections, as a higher proportion of men reported moderate/severe infections than women.

Some further limitations in the interpretation of our results should be acknowledged. First, our study was conducted on a large clinical cohort of over 2000 pwMS but was narrowed to our center’s regional impact area, so the generalizability of the observed patterns may be limited. Moreover, cultural characteristics within German populations may affect infection and vaccination patterns. For example, no infections were reported during the initial phases of the pandemic, which may be attributed to sociocultural factors rather than the inherent characteristics of pwMS. Second, our real-world analysis relied on anamnesis during medical visits and a retrospective analysis of medical reports, thus limiting our information gathering. We did not assess important social determinants of health such as education level, professional characteristics, alimentation, or social contacts, which could provide valuable information to our interpretation [40]. Also, we did not assess individual perceptions about the pandemic, which could help in interpreting the behavior of pwMS. Third, the lack of a regular standard method for data collection, such as a questionnaire, may leave room for the subjectivity of treating physicians and the compliance of patients, ultimately affecting the quality of data. Additionally, we used external data to confirm infections, but we did not conduct regular screening or testing at our center, which may have increased the risk of underreporting the results. However, we did not observe any significant patterns of missing values that might have biased the primary outcome measures. Given our study setting, we are confident that the underreporting of infections and vaccinations was more likely than the overreporting of them.

It is important to note that our real-world study was not designed to determine the efficacy or effectiveness of the vaccines under study. The number of vaccinations and infections (and their associations with sociodemographic and disease-related factors) are determined by several factors that have constantly changed and interacted during the diverse phases of the pandemic, such as the introduction of several variable nonpharmaceutical interventions, geographic regionality, the emergence of new viral variants of concern, various personal behavioral factors, and prioritization schemes for both primary vaccination series and booster doses. Since we could not adjust for these distorting factors in detail, our inferential analyses addressed only cross-sectional predictors and associations rather than temporal or causal relationships of the outcome measures.

## 5. Conclusions

Our study provides a unique analysis of real-world SARS-CoV-2 vaccinations and infection patterns in a large MS cohort. It may be beneficial to focus vaccination campaigns on young, active patients with low disability levels. In our research, the significance of DMT was unclear and did not play a central role. However, patients receiving B-cell depletion therapies should be given specific attention. Further research could shed light on the perception of the pandemic and social behavior of people with multiple sclerosis (pwMS), which could help us better understand this particular population.

## Figures and Tables

**Figure 1 vaccines-12-00265-f001:**
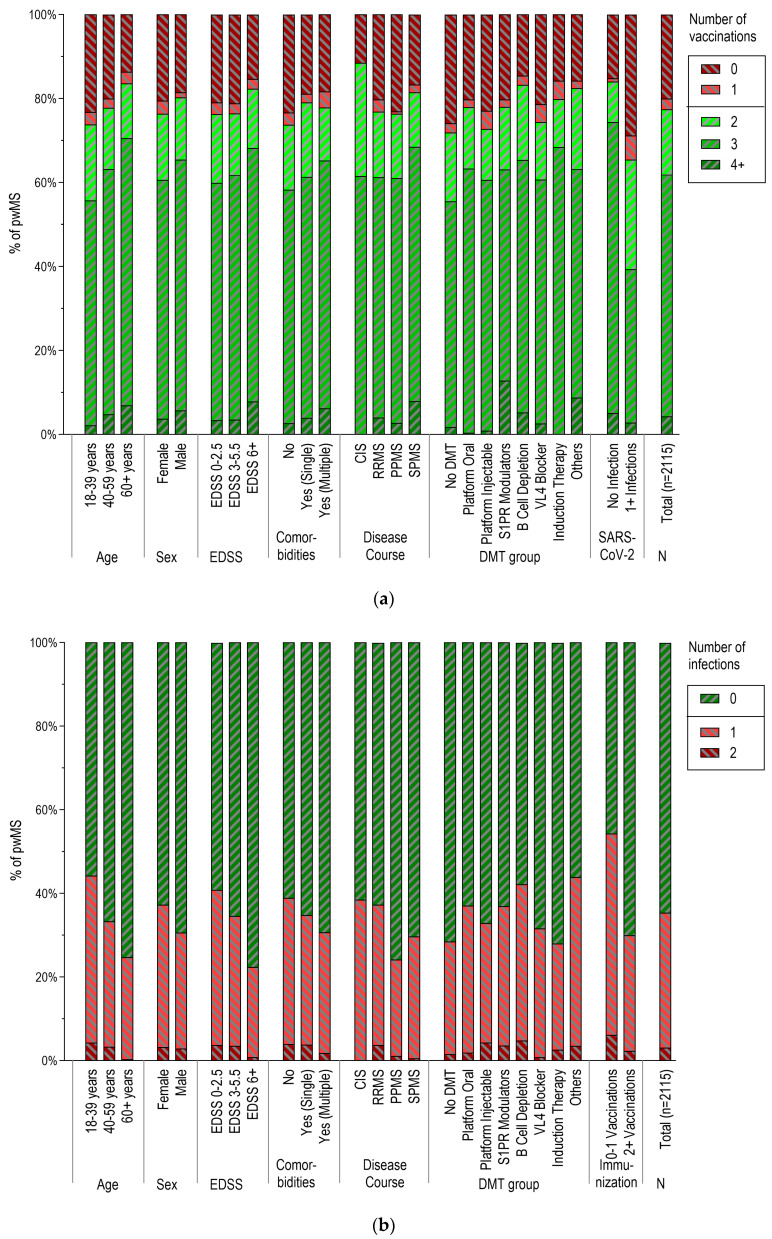
SARS-CoV-2 vaccination and infection patterns across demographic and disease-related subgroups (n = 2115). The distribution of relative frequencies (% of pwMS) across subgroup categories for (**a**) the number of vaccinations and (**b**) the number of infections is shown. In (**a**), the grouping of pwMS into those who received 0 or 1 vaccination is highlighted in red, and the proportion of pwMS with 2 or more vaccinations is summarized in green. In (**b**), the grouping of pwMS into those with 1 or more infections and those without infections is also represented in green and red colors, respectively. PwMS: people with multiple sclerosis. EDSS: expanded disability status scale. CIS: clinically isolated syndrome. RRMS: relapsing–remitting MS. PPMS: primary progressive MS. SPMS: secondary progressive MS. DMT: disease-modifying therapy.

**Figure 2 vaccines-12-00265-f002:**
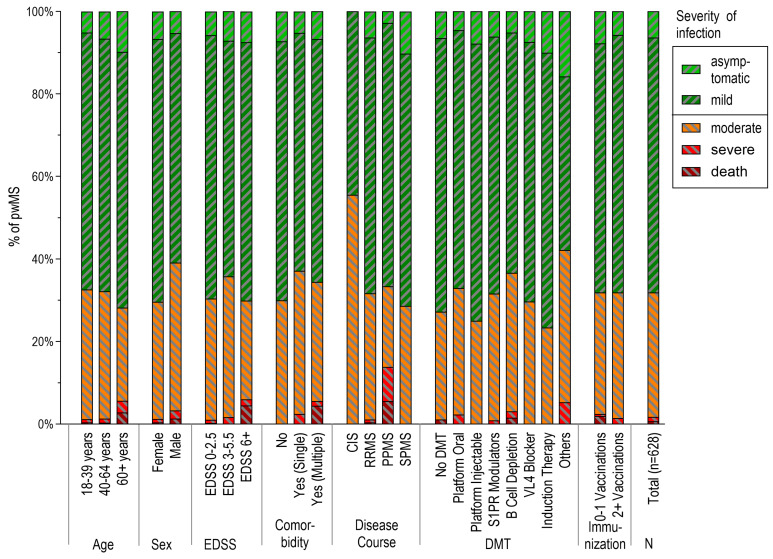
Severity of SARS-CoV-2 infections across demographic and disease-related subgroups (n = 628 pwMS). The distribution of the relative severity of infections (%) is shown. PwMS with asymptomatic or mild infections are highlighted in green, those with moderate infection are in yellow, and pwMS with severe infections or deathly cases are summarized in red. PwMS: people with multiple sclerosis. EDSS: expanded disability status scale. CIS: clinically isolated syndrome. RRMS: relapsing–remitting MS. PPMS: primary progressive MS. SPMS: secondary progressive MS. DMT: disease-modifying therapy.

**Figure 3 vaccines-12-00265-f003:**
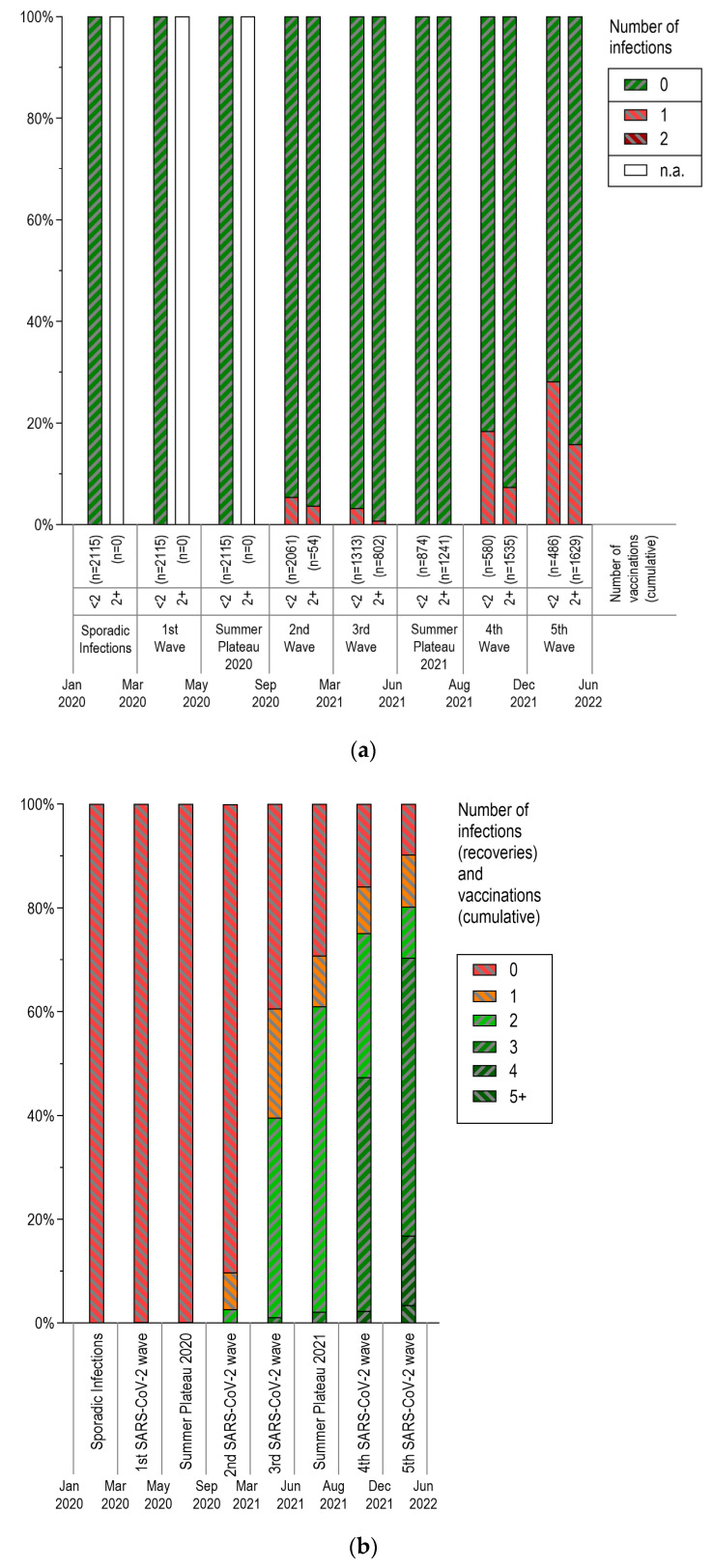
SARS-CoV-2 infections and vaccinations during the pandemic according to the phases defined by the Robert Koch Institute, Germany. (**a**) Relative number of SARS-CoV-2 infections (% of pwMS) over time, stratified by vaccination status. Very few cases were reported in the phases of sporadic cases and the first SARS-CoV-2 wave (one and three, respectively). The cumulative proportion of pwMS with immunological events (infections and vaccinations) over time is presented in (**b**). PwMS: people with multiple sclerosis. EDSS: expanded disability status scale. CIS: clinically isolated syndrome. RRMS: relapsing–remitting MS. PPMS: primary progressive MS. SPMS: secondary progressive MS. n.a.: not applicable.

**Table 1 vaccines-12-00265-t001:** Phases of the SARS-CoV-2 pandemic in Germany covered by the analysis.

Phase	Description	Period	Comment (Pandemic-Related Events)
1	Sporadic infections	27 January 2020–1 March 2020	- First confirmed SARS-CoV-2 infection in Germany (27 January 2020)
2	1st SARS-CoV-2 wave	2 March 2020–17 May 2020	- SARS-CoV-2 declared as global pandemic (11 March 2020) - First nationwide lockdown (22 March 2020)
3	Summer plateau 2020	18 May 2020–27 September 2020	
4	2nd SARS-CoV-2 wave	28 September 2020–28 February 2021	- Approval of first SARS-CoV-2 vaccine - Launch of vaccination campaign in Germany, with initial prioritization based on epidemiological and ethical criteria
5	3rd SARS-CoV-2 wave	1 March 2021–13 June 2021	- Alpha variant of concern
6	Summer plateau 2021	14 June 2021–1 August 2021	
7	4th SARS-CoV-2 wave	2 August 2021–26 December 2021	- Delta variant of concern - Recommendation of (first) booster vaccinations
8	5th SARS-CoV-2 wave	27 December 2021–30 May 2022	- Omicron variant of concern - Spreading far more easily than all earlier variants, including Delta

**Table 2 vaccines-12-00265-t002:** Demographic and Disease-specific Characteristics (*n* = 2115).

PwMS	
Sociodemographic characteristics:	
Age (years), mean (SD)	46.5 (±12.9)
18–39, *n* (%)	702 (33.2%)
40–59, *n* (%)	1005 (49.8%)
60+, *n* (%)	360 (17.0%)
Sex	
Female, *n* (%)	1533 (72.5%)
Body mass index (kg/m^2^), mean (SD)	24.5 (5.3)
Underweight (<18.5), *n* (%)	85 (4.0%)
Normal weight (18.5–< 25), *n* (%)	1060 (50.5%)
Overweight (25–< 30), *n* (%)	602 (28.7%)
Obesity (≥30), *n* (%)	353 (16.8%)
MS disease-specific characteristics:	
Disease duration, years, median [IQR]	10.0, [5–16]
<5 years, *n* (%)	601 (28.5%)
6–19 years, *n* (%)	1157 (54.9%)
≥20 years, *n* (%)	349 (16.6%)
MS disease course, *n* (%)	
RRMS	1691 (80.0%)
PPMS	182 (8.6%)
SPMS	216 (10.2%)
CIS	26 (1.2%)
EDSS, median [IQR]	2.5, [1.5–4.0]
0–2.5, *n* (%)	1094 (52.0%)
3–5.5, *n* (%)	624 (29.7%)
≥6, *n* (%)	384 (18.3%)
Comorbidities (*n* = 1927), mean (SD)	0.89 (0.88)
No (0), *n* (%)	723 (37.5%)
Single (1), *n* (%)	816 (42.3%)
Multiple (2+), *n* (%)	388 (20.1%)
DMT, *n* (%)	
No DMT	456 (21.6%)
Platform injectables	231 (10.9%)
Platform oral	267 (12.6%)
S1PR modulation	336 (15.9%)
B-cell depletion	537 (25.4%)
Induction therapies	114 (5.4%)
VCAM-1 blocker	117 (5.5%)
Others	57 (2.7%)
DMT treatment duration (years), mean (SD)	4.26 (3.67)

Note. PwMS: people with multiple sclerosis. RRMS: relapsing–remitting MS. PPMS: primary progressive MS. SPMS: secondary progressive MS. EDSS: expanded disability status scale. DMT: disease-modifying therapy.

**Table 3 vaccines-12-00265-t003:** SARS-CoV-2- and vaccination-related characteristics (n = 2115).

PwMS	
Number of SARS-CoV-2 infections, mean (SD)	0.39 (0.55)
0, n (%)	1365 (64.5%)
1, n (%)	684 (32.3%)
2, n (%)	66 (3.1%)
Severity of SARS-CoV-2 disease (n = 628), n (%)	
Asymptomatic	40 (6.4%)
Mild	388 (61.8%)
Moderate	189 (30.1%)
Severe	7 (1.1%)
Death	4 (0.6%)
Number of vaccinations, mean (SD)	2.25 (1.27)
0, n (%)	423 (20.0%)
1, n (%)	54 (2.6%)
2, n (%)	328 (15.5%)
3, n (%)	1220 (57.7%)
≥4 (%)	90 (4.3%)
Number of SARS-CoV-2 infections and/or vaccination, mean (SD)	2.64 (1.24)
0, n (%)	207 (9.8%)
1, n (%)	205 (9.7%)
2, n (%)	190 (9.0%)
3, n (%)	1131 (53.5%)
≥4, n (%)	382 (18.1%)

**Table 4 vaccines-12-00265-t004:** Factors associated with immunization and infection status during the SARS-CoV-2 pandemic (n = 2115 pwMS).

		(a) Vaccinations [≥2 vs. 0–1]				(b) SARS-CoV-2 Infections [≥1 vs. 0]			
		Univariable		Multivariable *		Univariable		Multivariable *	
		OR	95%-CI	OR	95%-CI	OR	95%-CI	OR	95%-CI
Age	cont. (years)	**1.016**	**1.008–1.024**	**1.016**	**1.008–1.024**	**0.969**	**(9.969–0.983)**	**0.975**	**0.968–0.982**
Age	18–39 years	Ref		Ref		Ref		Ref	
	40–59 years	1.243	0.995–1.553	1.235	0.988–1.542	**0.632**	**0.519–0.770**	**0.637**	**0.523–0.776**
	60+ years	**1.812**	**1.308–2.510**	**1.815**	**1.310–2.515**	**0.415**	**0.313–0.551**	**0.414**	**0.312–0.549**
Sex	Male	Ref.		Ref.		Ref.		Ref.	
	Female	0.797	0.629–1.008	0.793	0.626–1.005	**1.351**	**1.101–1.657**	**1.366**	**1.111–1.680**
EDSS	cont. (points)	**1.071**	**1.013–1.133**	1.016	0.952–1.085	**0.844**	**0.802–0.887**	**0.898**	**0.847–0.952**
EDSS	0–2.5	Ref		Ref		Ref		Ref	
	3–5.5	1.007	0.798–1.269	0.873	0.682–1.118	**0.766**	**0.625–0.940**	0.905	0.728–1.127
	6+	**1.441**	**1.071–1.939**	1.115	0.799–1.558	**0.418**	**0.319–0.546**	**0.563**	**0.417–0.759**
Comorbidity	No	Ref.		Ref.		Ref.		Ref.	
	Single	**1.345**	**1.062–1.703**	1.235	0.964–1.582	0.840	0.682–1.033	1.024	0.822–1.276
	Multiple	1.252	0.936–1.675	1.051	0.761–1.453	**0.694**	**0.535–0.905**	1.028	0.767–1.377
MS type	RRMS	Ref.		Ref.		Ref.		Ref.	
	PPMS	0.972	0.678–1.394	0.733	0.499–1.079	**0.534**	**0.375–0.761**	0.769	0.530–1.117
	SPMS	1.323	0.922–1.900	0.971	0.654–1.443	**0.706**	**0.518–0.960**	1.067	0.760–1.498
	CIS	2.306	0.689–7.721	2.352	0.701–7.894	1.047	0.472–2.322	1.026	0.460–2.288
Disease duration	cont. (years)	1.010	0.998–1.023	0.999	0.985–1.013	**0.981**	**0.971–0.992**	0.998	0.986–1.011
Disease duration	0–5 years	Ref.		Ref.		Ref.		Ref.	
	6–19 years	1.221	0.967–1.540	1.104	0.868–1.403	0.971	0.792–1.190	1.122	0.907–1.386
	20+ years	1.145	0.838–1.566	0.872	0.615–1.237	**0.603**	**0.451–0.805**	0.879	0.639–1.210
DMT	No DMT	Ref.		Ref.		Ref.		Ref.	
	Induction therapies	**1.972**	**1.181–3.293**	1.544	0.936–2.548	0.979	0.620–1.544	0.754	0.472–1.205
	Platform oral	**1.674**	**1.161–2.412**	1.376	0.966–1.960	**1.478**	**1.072–2.037**	1.216	0.873–1.694
	Platform injectables	1.242	0.863–1.786	1.041	0.730–1.483	1.230	0.874–1.730	1.021	0.719–1.450
	S1PR modulators	**1.601**	**1.140–2.250**	1.382	0.994–1.920	**1.467**	**1.086–1.981**	1.299	0.952–1.772
	B-cell depletion therapies	**2.357**	**1.711–3.247**	**1.938**	**1.429–2.629**	**1.836**	**1.408–2.395**	**1.552**	**1.175–2.052**
	VL4 blocker	1.623	1.000–2.634	1.132	0.713–1.797	1.160	0.747–1.800	0.776	0.490–1.228
	Others	**2.195**	**1.065–4.521**	1.834	0.900–3.740	**1.959**	**1.118–3.434**	1.709	0.963–3.303
Vaccinations	0–1 vaccination					Ref.		Ref.	
	2+ vaccinations					**0.360**	**0.292–0.444**	**0.378**	**0.306–0.466**
SARS-CoV-2 infections	0 (no infection)	Ref.		Ref.					
	1–2	**0.360**	**0.292–0.444**	**0.378**	**0.306–0.467**				

Note. Results of univariable and multivariable logistic analyses. Cross-sectional associations between SARS-CoV-2-related outcomes and sociodemographic and disease-related patient subgroups are presented. A descriptive presentation of the data accompanying the analyses is provided in Table 1 and Figure 1. * Adjusted for age and sex. Bold type: *p* < 0.05. OR: odds ratio; CI: confidence interval; Ref: reference category; cont.: independent variable treated as continuous; PwMS: people with MS; EDSS: expanded disability status scale; DMT: disease-modifying therapy; VCAM-1: Vascular Cellular Adhesion Molecule-1.

**Table 5 vaccines-12-00265-t005:** Correlations between the number of vaccinations, SARS-CoV-2 infections, and patient characteristics.

	Number of Infections	Number of Vaccinations	Age (years)
Number of infections	1	−0.314 **	−0.150 **
Number of vaccinations	−0.314 **	1	0.127 **
Age (years)	−0.150 **	0.127 **	1
Comorbidities (number)	−0.067 *	0.069 *	0.448 **
EDSS (points)	−0.135 **	0.082 **	0.547 **
Disease duration (years)	−0.057 **	0.089 **	0.456 **

Note. Cross-sectional correlations according to Spearman. Asterisks indicate the level of statistical significance (* *p* < 0.05, ** *p* < 0.001). EDSS: expanded disability status scale.

## Data Availability

Data are available upon reasonable request.

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
