# Peer review of "Insights from Real-World Practice: The Dynamics of SARS-CoV-2 Infections and Vaccinations in a Large German Multiple Sclerosis Cohort"

_vaccines, 2024, doi:10.3390/vaccines12030265_

Round 1

Reviewer 1 Report

Comments and Suggestions for Authors

In this manuscript the authors have studied a large cohort of patients with multiple sclerosis with respect to SARS-CoV-2 infections and vaccinations, from the beginning of the pandemic until May 2022. The methodologies are clearly described and appear to have been performed correctly. The results are carefully interpreted and discussed, and the limitations of the study identified. Overall, the work is interesting and is suitable for publication after correction of some issues indicated below.

 Minor:

 Abstract: The sentence “Higher 21 vaccination rates were associated with increased disability, older age, and comorbidities” could be misleading. Briefly clarify the probable explanation for this as provided in discussion section. Also in the Abstract, provide a last sentence to emphasize the relevance of the results.

Table 4, correct typo in multivariable in the third column.

Line 190: “were” instead of “where”

Line 244: Table e2?

Author Response

Dear reviewer,

we appreciate your feedback regarding the design and analysis of our study, as well as your general evaluation of our manuscript. Following, we address point-by-point your comments.

Comment 1: Abstract: The sentence “Higher 21 vaccination rates were associated with increased disability, older age, and comorbidities” could be misleading. Briefly clarify the probable explanation for this as provided in discussion section. Also in the Abstract, provide a last sentence to emphasize the relevance of the results.

  • We changed the formulation of that sentence and provided a brief explanation in the discussion. We added a sentence for a higher emphasis of the relevance of the results.

Comments 2 and 3: Table 4, correct typo in multivariable in the third column., Line 190: “were” instead of “where”

  • Thank you, we corrected the typos of table 4 and line 190.

Comment 4: Line 244: Table e2?

  • We corrected the mention of table e2 in line 244.

Best regards,

Hernan Inojosa

Reviewer 2 Report

Comments and Suggestions for Authors

The authors studied the effectiveness of vaccines against SARS-Cov-2 in preventing infections and the development of serious infections in a large patient population. The work is well-constructed and well-designed. I recommend that minor errors be corrected before publication.

Errors:

Table 2.

To explain the table, explain the abbreviations so that the table itself can be understood.

Table 4.

Explain the abbreviations in such a way that the table itself to be understandable.

Figure 2.

Space is missing (Severity of SARS-CoV-2infections). Please correct it.

Figure 3.

One miss type was detected (Dez); please change to “Dec”.

Comments:

Can you please explain why COVID-19 infection was not detected in pwMS patients during the period of sporadic COVID-19 infections, the first wave, and the summer plateau period?

Do you have data on post-vaccination immune response? Also, were there any differences in adverse reactions after vaccination and post-infection? The literature states that adverse reactions after SARS-CoV-2 vaccination are positively correlated with subsequent antibody titers (PMID: 36680026).

Have you had similar experiences? Could you comment on this opportunity?

Author Response

Dear reviewer,

we appreciate your feedback regarding the design and analysis of our study, as well as our manuscript. We further reviewed the presentation of our abstract and results for clearer understanding. Following, we provide a point-by-point response to your comments.

Comment 1: Table 2. To explain the table, explain the abbreviations so that the table itself can be understood.

-             We explained the abbreviations and classifications of table 2.

Comment 2: Table 4. Explain the abbreviations in such a way that the table itself to be understandable.

  • We explained the abbreviations and classifications of 4.

Comment 3: Figure 2. Space is missing (Severity of SARS-CoV-2infections). Please correct it.

-             We corrected the space missing in Figure 2.

Comment 4: Figure 3. One miss type was detected (Dez); please change to “Dec”.

  • We corrected the typing error of figure 3.

Comment 5: Can you please explain why COVID-19 infection was not detected in pwMS patients during the period of sporadic COVID-19 infections, the first wave, and the summer plateau period?

  • It is indeed remarkable, that there were few reported cases in our cohort at the immediate begin of the pandemic. The phases of the pandemic are based broader epidemiological data from the Robert Koch Institute for Germany and not a direct reflection of our specific setting. Data from our cohort may be influenced by the relatively low number of reported cases in the Saxony region, where the majority of our cohort is based. This could be attributed to sociocultural factors or chance rather than an inherent characteristic of the pwMS population. We added a comment of this fact/limitation in lines 367-369 of the discussion.

Comment 6: Do you have data on post-vaccination immune response? Also, were there any differences in adverse reactions after vaccination and post-infection? The literature states that adverse reactions after SARS-CoV-2 vaccination are positively correlated with subsequent antibody titers (PMID: 36680026).

Have you had similar experiences? Could you comment on this opportunity?

  • Investigating antibody titers corresponding to vaccinations and infections and their relationship with adverse reaction is in fact of great interest with possible relevance according to immune therapies. However, we did not address serological responses to these events in our cohort. In a different study of our group, we observed that patients under B-cell depletion produced a cellular-driven immune reaction to vaccinations although the humoral reaction was impaired. Still, we did not correlate these facts with adverse events.

Best regards,

Hernan Inojosa